# Detection of Soybean-Derived Components in Dairy Products Using Proofreading Enzyme-Mediated Probe Cleavage Coupled with Ladder-Shape Melting Temperature Isothermal Amplification (Proofman–LMTIA)

**DOI:** 10.3390/molecules28041685

**Published:** 2023-02-10

**Authors:** Fugang Xiao, Menglin Gu, Yaoxuan Zhang, Yaodong Xian, Yaotian Zheng, Yongqing Zhang, Juntao Sun, Changhe Ding, Guozhi Zhang, Deguo Wang

**Affiliations:** 1Henan Key Laboratory of Biomarker Based Rapid-Detection Technology for Food Safety, Food and Pharmacy College, Xuchang University, Xuchang 461000, China; 2College of Grain and Food, Henan University of Technology, Zhengzhou 450001, China

**Keywords:** food adulteration, dairy products, ladder-shape melting temperature isothermal amplification (LMTIA), soybean-derived components, proofreading enzyme-mediated probe cleavage (Proofman)

## Abstract

Food adulteration is a serious problem all over the world. Establishing an accurate, sensitive and fast detection method is an important part of identifying food adulteration. Herein, a sequence-specific ladder-shape melting temperature isothermal amplification (LMTIA) assay was reported to detect soybean-derived components using proofreading enzyme-mediated probe cleavage (named Proofman), which could realize real-time and visual detection without uncapping. The results showed that, under the optimal temperature of 57 °C, the established Proofman–LMTIA method for the detection of soybean-derived components in dairy products was sensitive to 1 pg/μL, with strong specificity, and could distinguish soybean genes from those of beef, mutton, sunflower, corn, walnut, etc. The established Proofman–LMTIA detection method was applied to the detection of actual samples of cow milk and goat milk. The results showed that the method was accurate, stable and reliable, and the detection results were not affected by a complex matrix without false positives or false negatives. It was proved that the method could be used for the detection and identification of soybean-derived components in actual dairy products samples.

## 1. Introduction

In recent years, with the rapid development in China’s economy and the improvement in people’s living standards, the demands for food quality and safety have also improved. At the same time, dairy products as a kind of food with high nutritional value have become an indispensable food in the diet structure of residents in our country. With the increase in people’s income and change in consumption concept, consumers in the dairy market continue to pursue healthy, high-quality and safe products, and promote the continuous upgrading of dairy consumption structure [1]. According to the data from 2005 to 2017, the data envelopment analysis (DEA) model was used to analyze the quality of dairy products in different importing countries due to different production technologies [2], and there was the problem that importing countries were too concentrated [3]. Therefore, it is necessary to establish an efficient and rapid detection technology for dairy products.

Driven by profits, fraud such as shoddy and adulterated dairy products with high value often occurs [4,5,6]. At present, the main adulterants in dairy products are nitrogen-containing compounds, vegetable proteins (such as soy protein) and animal proteins. Water, rice soup, hydrolyzed leather protein, plant fat end, soy protein, whey powder, melamine and urea are usually added to dairy products [7]. After the “Big Head Baby” scandal of Sanlu cow milk powder [8], with the efforts of many scientific researchers, the detection methods of melamine and other nonprotein nitrogen content have become more and more advanced, while profitmaking through the adulteration of nonprotein nitrogen content has become a thing of the past. However, low-cost plant-derived proteins have been used to replace the nonprotein nitrogen, and the main plant-derived proteins are soybean proteins [9]. Therefore, in the process of quality control, which is particularly important for government supervision and enterprise production, it is necessary to use fast and reliable detection methods to monitor raw materials.

Due to the continuous improvement in adulteration technology, it is difficult to identify the authenticities in dairy products using ordinary sensory methods or simple chemical methods [10,11,12,13]. DNA-based detection methods are highly sensitive and repeatable, and have been widely used in the adulteration of dairy products [14].

The nucleic acid detection method has strong specificity, high sensitivity and accurate results [15] and is suitable for species identification [16]. The existing nucleic acid amplification methods can be divided into two categories: thermal cycle amplification and isothermal amplification [17]. The commonly used thermal cycle amplification methods are polymerase chain reaction (PCR) and ligase chain reaction (LCR) [18,19,20]. The limitation of these methods is that they need a thermal cycle instrument. Isothermal amplification technology is developed on the basis of thermal cycle amplification technology. Isothermal amplification does not need a thermal circulator and is simpler and shorter than variable-temperature amplification. There are many technologies that have been developed, such as nucleic acid sequence-based amplification (NASBA) [21], self-sustained sequence replication (3SR) [22], helicase-dependent isothermal DNA amplification (HDA) [23], exponential amplification reaction (EXPAR) [24], strand displacement amplification (SDA) [25], recombinase polymerase amplification (RPA) [26], cross-primer amplification (CPA), rolling circle amplification (RCA) [27], loop-mediated isothermal amplification (LAMP) [28]. However, these methods still have different disadvantages such as a long amplification time and nonspecific amplification [28,29].

The DNA mentioned in the LAMP principle is in dynamic equilibrium at about 65 °C. When any primer extends to the complementary part of the double-stranded DNA for base pairing, the other strand will dissociate and become a single strand. If this theory is established, one pair of primers can also be amplified, so there is no need for high-temperature denaturation in the PCR reaction. Adding helicase to the constant-temperature amplification technology of DNA, dependent on helicase to unlock the double strand of DNA, will also be superfluous. Our team has proved that this theory is wrong [30,31]. However, in many literature reports, based on this theoretical basis, the reaction was realized through LAMP. Our team analyzed this and believed that the single strand of DNA was generated due to the influence of heating or chemical factors during the preparation of the DNA template, and the amplification was realized, which further increased the chance and uncertainty of this amplification. If the 3–4 bases at the 3′ end of any two primers of LAMP have two complementary sequences on the same primer, there must be nonspecific amplification under the common LAMP reaction system and reaction conditions. The design and screening of LAMP primers should avoid this situation, but the mechanism of this nonspecific amplification needs further study. Taking the specific gene *inv*A of *Salmonella* as the target sequence, a set of LAMP primers was designed and screened, which did not exist in this case, and the negative control did not have nonspecific amplification. In short, the false positive in the LAMP reaction is mainly caused by nonspecific amplification between primers; thus, primer design and screening are crucial [30,31]. Based on the above analysis, with reference to denaturing gradient gel electrophoresis (DGGE), our team improved the method of micro-denaturing partial chain breaking and independently developed a new isothermal amplification technology with only one pair of primers: ladder-shape melting temperature isothermal amplification (LMTIA) [32,33]. Its amplification can be divided into two major categories, namely, herringbone type and nonherringbone type, and subdivided into eight subcategories according to the melting temperature curve and primer type. The LMTIA reaction is divided into the initial structure formation stage and the exponential amplification stage. The LMTIA technology uses one pair of primers or two pairs of nested primers and a thermostable DNA polymerase (large fragment) to amplify the internal transcribed spacer (ITS) of rice in a ladder melting curve. Compared with the LAMP method with the same level of specificity, the sensitivity of the nested-primer LMTIA method is increased by 50 times. The primer design method of LMTIA is simple, and the amplification efficiency of LMTIA is high and fast. The amplification reaction enters the platform stage from the exponential stage in about 20 min.

The LMTIA technology makes the thermal cycler no longer a necessary instrument for nucleic acid amplification technology [34]. The LMTIA technology has the following advantages [32]: 

(1) the reaction mechanism is simple and it does not need thermal denaturation, nor does it need the synergy of other enzymes except Bst DNA polymerase; 

(2) the primer design is simple, and the LMTIA primers can be designed using common PCR primer software; 

(3) the reaction time is short, and the amplification reaction time is less than 30 min; 

(4) the specificity is strong, because the LMTIA reaction requires not only the complementary pairing of the primer and the target sequence, but also the ladder structure of the melting temperature curve of the target sequence; 

(5) it has high sensitivity, reported to be more than 50 times higher than the LAMP; 

(6) low requirements for the length of the target sequence. LMTIA can amplify more than 60 nt of the target sequence, while LAMP requires the length of the target sequence to be at least 200 nt; 

(7) it can be used in a wide range of applications, including in double-stranded DNA as a template or in single-stranded DNA or RNA as a template, and does not require reverse transcriptase [35]. 

Compared with the traditional PCR technology, it is accurate for the objective and it has a relatively low detection limit and short amplification time. After the amplification, the fluorescence generation can be directly observed. Currently, it is widely used in the detection of food adulteration [36,37]. A rapid method for the detection of dairy soybean genes was established, which can be used for the primary screening of dairy products.

Probes based on enzymatic cleavage have a wide range of applications in molecular biology, but their preparation is complex and requires the incorporation of tetrahydrofuran base-free site analogues. Chen et al. cleaved the enzyme mediated probe (Proofman) and developed recombinase-assisted loop-mediated isothermal amplification (RALA) [38,39]. A novel nucleic acid sequence-specific detection platform for soybean was developed using LMTIA coupled with a Proofman probe. The Proofman-based LMTIA was proved to be an attractive option for accurately diagnosing soybean. In this study, the nucleic acid sequence of soybean (*Glycine max* (Linn.) Merr.) endogenous gene was selected to design the LMTIA specific primers, and a specific, rapid and accurate LMTIA-coupled Proofman detection method was established to detect soybean-derived components in dairy products.

## 2. Results

### 2.1. LMTIA Primer Design

The primer sequences are shown in Table 1. Then, using the Proofman probe, labeled with a fluorophore and quencher at the 3′ end and 5′ end, respectively, the sequence-specific detection of LMTIA products was achieved through the binding and cleavage of the Proofman probe, thus enhancing the accuracy of the LMTIA reaction.

### 2.2. Optimization of the Proofman–LMTIA Reaction Temperature

The Proofman–LMTIA assay was conducted at 55 °C, 56 °C, 57 °C and 58 °C, and the results are shown in Figure 1. The negative control DEPC water was not amplified, while the soybean gDNA was amplified and the amplification effect was good. Soybean primers were not significantly affected by temperature change and began to amplify at around 12 cycles, but from the perspective of fluorescence intensity and repeatability, 57 °C was the best, so it was determined as the best reaction temperature.

### 2.3. Specificity of the Proofman–LMTIA Assay

In order to ensure the unique specificity of the designed primers for soybean, the optimized Proofman–LMTIA reaction system was used to perform a Proofman–LMTIA test on the gDNA grouping of soybean, sesame, sunflower, walnut, sheep, cattle and others, with DEPC water as the negative control and soybean gDNA as the positive control. The result is shown in Figure 2. Under the condition of 57 °C, soybean DNA was amplified normally, while DEPC water and other control DNA were not amplified. Therefore, the Proofman–LMTIA assay was of high specificity and only amplified the soybean DNA, and the primers designed in this experiment had good specificity and could meet the screening of soybean-derived components in dairy products.

### 2.4. Sensitivity of the Proofman–LMTIA Assay

In order to ensure the sensitivity of the experimental method, the sensitivity of the designed soybean primer was tested. The cryopreserved soybean DNA was diluted to three concentration gradients of 100 pg/μL, 10 pg/μL and 1 pg/μL, respectively, and the Proofman–LMTIA assay was performed at 57 °C with DEPC water as the negative control and soybean DNA as the positive control. The amplification curve is shown in Figure 3. Soybean DNA at concentrations of 100 pg/μL, 10 pg/μL and 1 pg/μL were all well amplified, but the stability and repeatability were not good with 1 pg/μL. Therefore, the concentration of soybean DNA detected using the Proofman–LMTIA assay was 1 pg/μL.

### 2.5. Sample Testing

In order to verify whether the system could detect soybean-derived components in dairy products, Mengniu pure cow milk and Ama-jia pure cow milk of the Qinghai-Tibet Plateau were tested, respectively, under the soybean reaction system and the bovine reaction system. In the bovine reaction system, DEPC water was the negative control and bovine DNA was the positive control. Both cow milk and cow DNA were amplified. In the soybean reaction system, DEPC water was also used as the negative control and soybean DNA was used as the positive control. The cow milk DNA was unamplified, while the soy DNA was amplified. The results are shown in Figure 4 and Figure 5. The same method was used to detect Jomilk selected pure goat milk, and the results are shown in Figure 6. In the sheep system, the DNA from both sheep milk and goat milk was amplified. In the soy system, only soy DNA was amplified.

## 3. Material and Methods

### 3.1. Target Sequence Selection and LMTIA Primer Design

There are three criteria for the target sequence selection, as described by Wang et al. [32]: the melting temperature curve of the sequence is of a ladder type; the GC content of the sequence is generally 40–80%; and the sequence has high specificity. The sequence with a ladder-shaped melting temperature curve was selected as a target from the internal transcribed spacer (ITS) gene of soybean using the software Oligo 7 (Molecular Biology Insights, Inc. Colorado Springs, CO, USA) [32]. The LMTIA primers were designed with the online software Primer3Plus (http://www.primer3plus.com, accessed on 20 March 2022), and the parameters of primers were set as described in a previous report [32]. The Proofman probes were designed based on the primer LB sequence, with the fluorophore and the quencher labeled at the end of the 3’ end mismatch nucleotide and 5’ end nucleotide, respectively [39].

### 3.2. DNA Extraction

When extracting DNA from cow milk and goat milk, DNA enrichment is required first, and the specific steps are as follows: 10 mL milk samples were collected in a centrifugal tube and centrifuged at 4 °C at 7000 r/min for 10 min. Then, 600 μL of PBS was added to the bottom precipitate, and the precipitate was suspended by repeated blowing and transferred to a 2 mL centrifuge tube. Then, the precipitate was put into a high-speed refrigerated centrifuge and centrifuged at 4 °C and 12,000 r/min for 10 min; the supernatant was poured out; and the bottom precipitate was removed. After the bottom precipitation was weighed, DNA was extracted from the corresponding milk samples using a food DNA extraction kit (TIANGEN DP326, Tiangen Biotech [Beijing] Co., Ltd., Beijing, China). After the purity and concentration were determined, the samples were frozen at 4 °C.

A plant genome kit (TIANGEN DP305, Tiangen Biotech [Beijing] Co., Ltd., Beijing, China) was used to extract DNA from soybean, sunflower, sesame, walnut and other plants, and an animal genome kit (TIANGEN DP304, Tiangen Biotech [Beijing] Co., Ltd., Beijing, China) was used to extract DNA from beef and mutton. After the DNA purity and concentration were determined, the DNA was stored at 4 °C.

### 3.3. Proofman–LMTIA Reaction

For the Proofman–LMTIA reaction, 10 μL of the reaction system (0.06 μL of primer F and primer B, 0.02 μL of primer LF and primer LB, 0.014 μL of Pr, 4 μL of the premix of dNTP and Bst DNA Polymerase (Merit Biotech [Shandong] Co., Ltd., Heze, China), and 2 μL of DNA template) was added for the conventional fluorescent reaction. As for Proofman, instead of SYBR Green I, 0.014 μL of the Proofman probe was added.

### 3.4. Proofman–LMTIA Reaction Temperature Optimization

According to the reaction system described in 3.3, the Stepone plus real-time PCR system (Applied Biosystems, Thermo Fisher Scientific, Foster City, CA, USA) reaction temperatures were set as 55 °C, 56 °C, 57 °C and 58 °C, respectively, with heating for 60 min (90 s for 1 cycle; the purpose of the setting cycle was 90 s for the one-time acquisition of the fluorescence signal and FAM was selected for the fluorescence channel) [40]. The positive control was soybean genomic DNA (gDNA), and the negative control was DEPC water. After the amplification, the amplification curve of real-time fluorescent quantitative Proofman–LMTIA was carefully observed and studied to determine the optimal reaction temperature of soybean primers.

### 3.5. Specificity Determination of Proofman–LMTIA Assay

With H_2_O as the negative control and 1 ng/μL soybean gDNA as a positive control, the specificity of the established Proofman–LMTIA assay was tested with the gDNAs of beef, mutton, sunflower, corn, walnuts and others. A 10 μL reaction system described in 3.3 was applied, and each sample was repeated twice (repeated thrice for soybean gDNA) in the Archimed time-resolved fluorescence quantitative PCR system (RocGene [Beijing] Technology Co., Ltd., Beijing, China).

### 3.6. Sensitivity Determination of Proofman–LMTIA Assay

Soybean gDNA was diluted to 100 pg/μL, 10 pg/μL and 1 pg/μL, and the sensitivity was tested under the optimal temperature reaction conditions. A 10 μL reaction system described in 3.3 was applied, and two parallel samples were set for each sample.

### 3.7. Actual Sample Testing

The gDNA of a brand of cow milk and goat milk was extracted and verified using the previously established Proofman–LMTIA method of beef and mutton [40,41]. Then, the established soybean Proofman–LMTIA detection system was used to detect whether soybean-derived components were present. DEPC water was the negative control and beef gDNA was the positive control; in the sheep reaction system, DEPC water was the negative control and mutton gDNA was the positive control; and in the soybean reaction system, DEPC water was the negative control and soybean gDNA was the positive control. A reaction system in an amount of 10 μL was used with at least two parallelisms for each sample.

The flow chart of Proofman–LMTIA is shown in Figure 7.

## 4. Discussion

The separation and extraction of high-quality DNA is the key to genetic testing. DNA extracted from dairy products is mainly obtained from somatic cells, but there are only 20,000 to 200,000 individual cells per milliliter of milk, and the DNA content is not high [42]. In addition, dairy products need to go through spray drying, high-temperature sterilization and other processing in the production process, which will lead to DNA breakage or degradation, exacerbating the difficulty of DNA extraction. In this study, DNA extraction kits suitable for dairy products and other different samples were selected through comparative tests; the amplified DNA from dairy products was successfully extracted using nucleic acid enrichment and a food DNA extraction kit.

As reported in the previous paper [32], the LMTIA technique based on LAMP can be used under isothermal conditions to detect *Oryza sativa* L. DNA with 50 times more sensitivity than LAMP. The sequence with a ladder-shaped melting temperature curve was selected as a target from the internal transcribed spacer (ITS) gene of soybean using the software Oligo 7. Based on the selected sequence, LMTIA primers were designed, and meanwhile, the Proofman probes were also designed based on the primer LB sequence, with the fluorophore and the quencher labeled at the end of the 3’ end mismatch nucleotide and the 5′ end nucleotide, respectively. Further, the primers displayed high specificity at 57 °C for the soybean DNA (Figure 1). This simple and rapid method could be applied to determine the soybean-derived components in dairy products.

In recent years, food fraud has become an urgent problem linked to the traceability of food products, and food authenticity has become increasingly important as a result of food adulteration [37]. Dairy products are one of the important sources of nutrition for the human body. However, fraud—driven by profits—often occurs, such as shoddy and adulterated dairy products with high value [4,5,6]. Therefore, the development of selective and sensitive detection techniques is a key challenge for the detection of authentic food, especially dairy products. DNA-based detection methods are highly sensitive and repeatable, and have been widely used in the adulteration of food products [14]. Zhu et al. (2019) [36] reported that the LAMP method could be used to detect pork components in common meat products, and the detection limit was 10 pg/uL. The LMTIA assay was also a reliable method for the rapid detection of cassava components in sweet potato starch noodles and could specifically distinguish a 0.01% (*w*/*w*) cassava component added into sweet potato starch [37]. Wang et al. (2022) [40] developed the LMTIA method for the detection of duck adulteration in beef with a 0.1% limit of detection. The established Proofman–LMTIA method for the detection of soybean-derived components in dairy products was sensitive to 1 pg/μL (Figure 3). Therefore, the DNA-based detection methods can be applied in the detection of authentic food, as well as the preliminary screening of food products on the market. In addition, with the advantages of simple equipment requirements, simple operation, high detection efficiency, rapid detection, strong specificity and high sensitivity, it is suggested to continuously strengthen the integrity and application of Proofman–LMTIA technology, and explore its application in food adulteration detection, pathogenic bacteria detection, virus detection and species relationship detection.

In this study, with a Proofman probe instead of SYBR Green I, multiplex detection can be achieved in one pot, improving the accuracy of detection. Multiple probes can be added into the system to realize multiple detection, which is suitable for the rapid detection of food adulteration.

The LMTIA method requires not only the complementary pairing of the primer and target sequence, but also the ladder-type melting curve of the target sequence to provide a single-chain template for the amplification reaction [32]. Therefore, in theory, the LMTIA method should have high specificity. In this study, the established Proofman–LMTIA method was used to detect the gDNA grouping of sesame, sunflower, walnut, sheep, cattle and others, with DEPC water as the negative control, which was negative, and the detected gDNA of soybean as the positive control, which preliminarily confirmed the specificity of the Proofman–LMTIA method.

On the basis of previous research, further research was carried out to clarify the mechanism of template unwinding in LAMP technology through theoretical analysis and practical verification, and to use a pair of primers and other technical means to solve the false-positive problem caused by nonspecific amplification and apply it to the LMTIA technology. The primer design screening method was established using mathematical induction, and the LMTIA technology was improved by combining a fluorescent probe and a catalytic enzyme to carry out the qualitative and quantitative detection of the authenticity of dairy products so that its sensitivity reached the same level as that of fluorescent PCR; the detection time was shortened to 20 min so as to further improve the LMTIA technology. This study can further improve the theory of nucleic acid detection, break through the bottleneck of LMTIA technology and promote the development and application of new rapid diagnoses of epidemic diseases and rapid detection kits for food safety.

## 5. Conclusions

LMTIA primers were designed for the soybean ITS gene, and a method of Proofman–LMTIA detection of soybean-derived components in dairy products was developed. The specific test showed that the primers could distinguish soybean DNA from the DNA of sesame, sunflower, walnut, mutton and beef. The optimal reaction temperature of the established soybean Proofman–LMTIA assay was 57 °C, and the sensitivity was up to 1 pg/μL at this temperature. The Proofman–LMTIA method established in this study has the advantages of simple equipment requirements, simple operation, high detection efficiency, rapid detection, high sensitivity and strong specificity, and can be applied to the detection of soybean-derived components in dairy products.

## Figures and Tables

**Figure 1 molecules-28-01685-f001:**
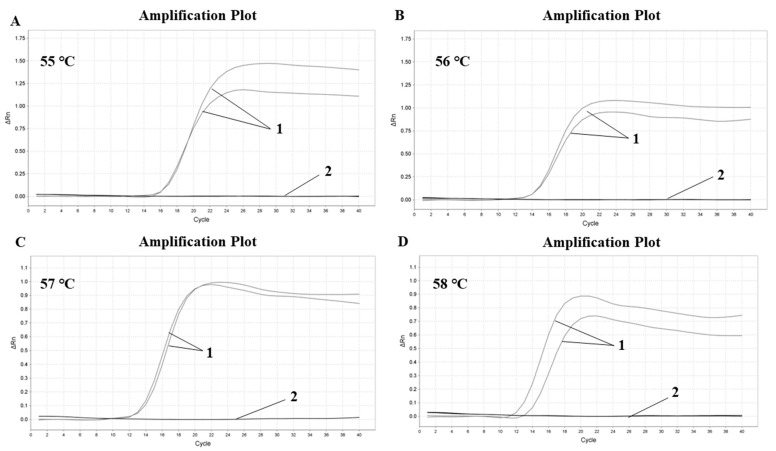
Optimization of Proofman–LMTIA reaction temperature for ITS gene in soybean. (**A**) 55 °C, (**B**) 56 °C, (**C**) 57 °C, (**D**) 58 °C. 1: positive controls (10 ng of gDNA from soybean); 2: negative controls (ddH_2_O).

**Figure 2 molecules-28-01685-f002:**
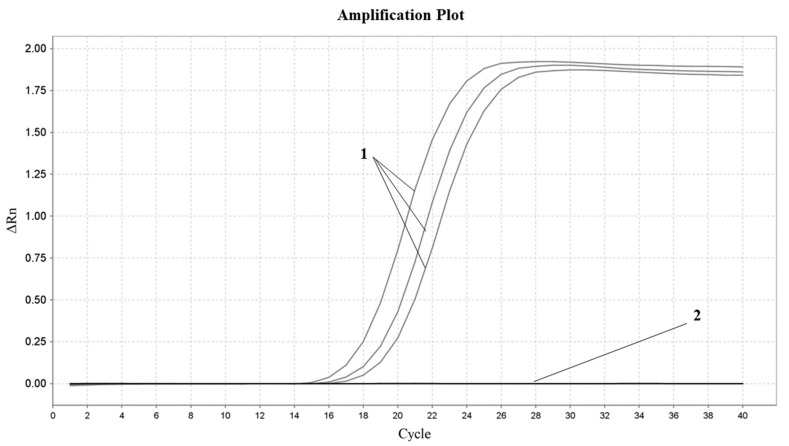
Proofman–LMTIA response specificity assay of ITS gene in soybean. 1: Soybean gDNA; 2: ddH_2_O, sesame, sunflower, walnut, sheep, cattle and others.

**Figure 3 molecules-28-01685-f003:**
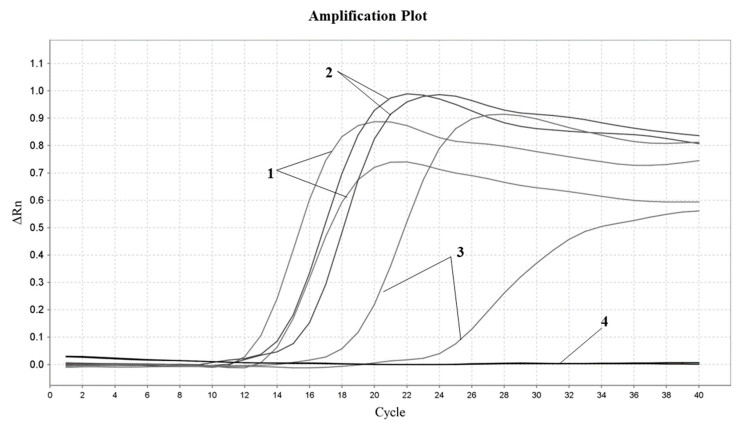
Proofman–LMTIA response sensitivity test for ITS gene in soybean. Varying amounts of gDNAs of the following: 1: soybean 100 pg/μL; 2: soybean 10 pg/μL; 3: soybean 1 pg/μL; 4: ddH_2_O.

**Figure 4 molecules-28-01685-f004:**
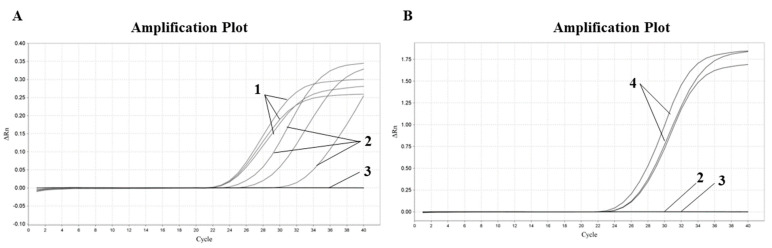
Proofman–LMTIA method for testing Mengniu pure cow milk. (**A**): Verification of cow milk DNA: 1: beef gDNA; 2: cow milk DNA; 3: ddH_2_0; (**B**): Detection of soybean-derived components in cow milk: 2: cow milk DNA; 3: ddH_2_0; 4: soybean gDNA.

**Figure 5 molecules-28-01685-f005:**
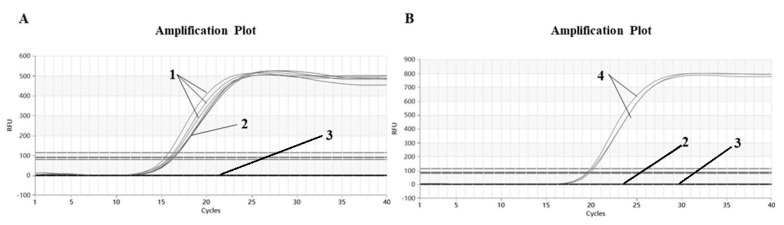
Proofman–LMTIA method for testing Ama’s pure cow milk in Qinghai–Tibet Plateau. (**A**): Verification of cow milk DNA: 1: beef gDNA; 2: cow milk DNA; 3: ddH_2_0; (**B**): Detection of soybean-derived components in cow milk: 2: cow milk DNA; 3: ddH_2_0; 4: soybean gDNA.

**Figure 6 molecules-28-01685-f006:**
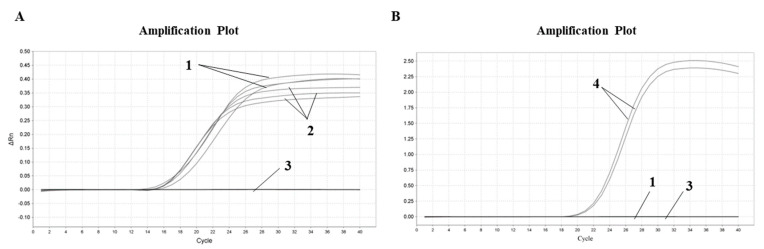
Proofman–LMTIA method for detection of Jomilk selected pure goat milk. (**A**): Verification of goat milk DNA: 1: goat milk DNA; 2: mutton gDNA; 3: ddH_2_0; (**B**): Detection of soybean-derived components in goat milk: 1: goat milk DNA; 3: ddH_2_0; 4: soybean gDNA.

**Figure 7 molecules-28-01685-f007:**
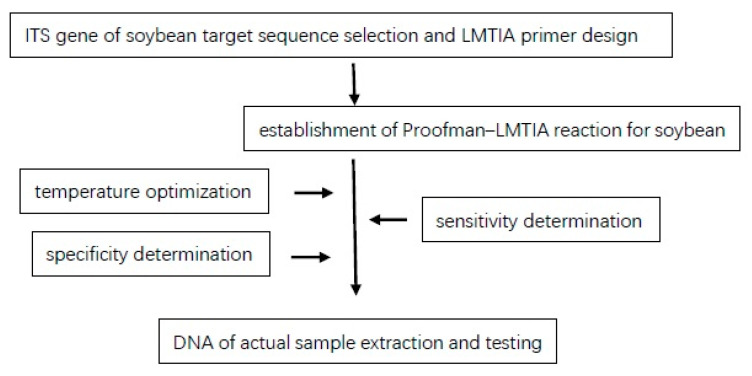
Flow chart of Proofman–LMTIA.

**Table 1 molecules-28-01685-t001:** LMTIA primer sequences for ITS gene in soybean.

Primer	Sequence (5′–3′)
F	CGTGCACGCAAAGGGTTTTTCCACGCTCGAGACCAATCAC
B	TGCACGCACGCTCCCTTTTATGCTTAAACTCAGCGGGTAG
LF	TCCAGAACTGACCGGCTCGCA
LB	ACGAGACCTCAGGTCAGGCG
Pr	BHQ 1-ACGAGACCTCAGGTCAGGCG-FAM

## Data Availability

The data presented in this study are available on request from the corresponding author.

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
