# Peer review of "Detection of Soybean-Derived Components in Dairy Products Using Proofreading Enzyme-Mediated Probe Cleavage Coupled with Ladder-Shape Melting Temperature Isothermal Amplification (Proofman–LMTIA)"

_molecules, 2023, doi:10.3390/molecules28041685_

Round 1
Reviewer 1 Report
This is an interesting study in which the LMTIA method was used to detect the soybean ITS gene in dairy products as proof of adulteration. Food adulteration is a global economic and health concern and there is a growing need for development of rapid and accurate detection methods.
Some suggestions for improvement:
Line 35: “development of our economy”, please specify which economy
Line 40: “which promotes”, which refers to what?, it is not clear, please rephrase
Line 41: “the DEA model” please also state the analytical description of the abbreviation
Line 54: this sentence needs a reference
Line 63: “Compared with the traditional PCR technology, it is accurate for the object, relatively low detection limit and short amplification time”, change to “Compared with the traditional PCR technology, it is accurate for the object and it has relatively low detection limit and short amplification time”. Also, perhaps you should change “it is accurate for the object” because it is not very clear as a meaning
Line 76: please also use the scientific name of soybean
Line 96: please provide name of kit
Line 136: the section 3.1. “LMTIA primer design” and Table 1 should be placed in the Materials and Methods
Line 151: “soybean DNA was amplified and well amplified” please rephrase
Line 188: the bovine reaction system has not been mentioned in the “materials and methods” section
Line 187: it should be mentioned if these milks are both cow’s milk
Line 193: please explain “jomilk”
Line 197: the sheep system is also not mentioned in the “materials and methods” section
Line 216: “200,000 to 200,000 individual cells” please correct
Reviewer 2 Report
Dear author(s):
Detection of soybean derived components in dairy products using proofreading enzyme-mediated probe cleavage coupled with ladder-shape melting temperature isothermal amplification (Proofman-LMTIA)
After an exhaustive revision, the manuscript is Reconsider after major revision (control missing in some experiments). In general, the study is closely connected to the journal's objectives. The study is very interesting. The English is good. The introduction is good, the materials and methods need a Figure. The results need numerical description and the discussions need a lot of change, since it need explication of the results, comparison with other studies, and explication (discussion) of the results obtained with respect to other studies, the authors only write introduction and conclusions in discussion section.
In the following pages, I give a detailed revision of the manuscript.
ABSTRACT
The abstract is good.
1. INTRODUCTION
The introduction is very clear, concise and precise, with good English, and it has updated references.
2. MATERIALS AND METHODS
General comments
This section is clear, very detail and complete. The English is good. The authors must add a Figure that represents all the complete methodology. This Figure will help to understand the methodology. Some observations:
2.2. DNA extraction
2.3. Proofman-LMTIA reaction
2.4. Proofman-LMTIA reaction temperature optimization
What is the reference?
3. RESULTS
The section of "Results" is characterized by a very detailed description of the results.
3.1. LMTIA primer design
The authors need to add more information on results (Table 1).
3.2. Optimization of the Proofman-LMTIA reaction temperature
3.3. Specificity of the Proofman-LMTIA assay
3.4. Sensitivity of the Proofman-LMTIA assay
3.4. Sensitivity of the Proofman-LMTIA assay
For all the points, the authors need to add numerical description.
4. DISCUSSION
The section of "Discussion" is characterized by the explication of the results, comparison with other studies, and explication (discussion) of the results obtained with respect to other studies.
Lines 209-213. These lines are unnecessary, since these lines are an introduction.
Lines 214-221. These lines are unnecessary, since these lines are an introduction, with a little conclusion.
Lines 222-225. These lines are unnecessary, since these lines are a little conclusion.
Lines 226-241. These lines are unnecessary, since these lines are an introduction and conclusions.
4. CONCLUSIONS
It is good.

Reviewer 3 Report
The current manuscript reports the development of a proofreading enzyme-mediated probe cleavage coupled with ladder-shape melting temperature isothermal amplification (Proofman-LMTIA) for the detection of soybean derived components in dairy products.
In general, this is an important and interesting work. I have, however a few comments or suggestions.
1. Figure 4, “2” and “4” should be revised.
2. Subsection 3.5 should be at the beginning of Materials and methods.
3. In" Specificity of the Proofman-LMTIA assay", the optimized Proofman-LMTIA reaction system was used to perform Proofman-LMTIA test on the genomic DNA grouping of sesame, sunflower, walnut, It was unclear why they chose these targets to test for specificity?How was the DNA concentration determined?
4. What is the detection limit?
5. To determining adulteration, wouldn't a quantitative approach be more appropriate?
Round 2
Reviewer 1 Report
The manuscript can be published in the present form
Reviewer 2 Report
Dear Author(s)
After an exhaustive revision, the manuscript is Accept in present form. The resubmitted manuscript has been completely improved compared to its previous version. Therefore, the manuscript can be published in “Molecules”.
Best regards